# Identification of Candidate Ice Nucleation Activity (INA) Genes in *Fusarium avenaceum* by Combining Phenotypic Characterization with Comparative Genomics and Transcriptomics

**DOI:** 10.3390/jof8090958

**Published:** 2022-09-13

**Authors:** Shu Yang, Mariah Rojas, Jeffrey J. Coleman, Boris A. Vinatzer

**Affiliations:** 1School of Plant and Environmental Sciences, Virginia Tech, Blacksburg, VA 24061, USA; 2College of Advanced Agricultural Sciences, Zhejiang A&F University, Hangzhou 311300, China; 3Department of Entomology and Plant Pathology, Auburn University, Auburn, AL 36849, USA

**Keywords:** ice nucleation activity, *Fusarium avenaceum*, fungi, comparative genomics, transcriptomics

## Abstract

Ice nucleation activity (INA) is the capacity of certain particles to catalyze ice formation at temperatures higher than the temperature at which pure water freezes. INA impacts the ratio of liquid to frozen cloud droplets and, therefore, the formation of precipitation and Earth’s radiative balance. Some *Fusarium* strains secrete ice-nucleating particles (INPs); they travel through the atmosphere and may thus contribute to these atmospheric processes. *Fusarium* INPs were previously found to consist of proteinaceous aggregates. Here, we determined that in *F. avenaceum*, the proteins forming these aggregates are smaller than 5 nm and INA is higher after growth at low temperatures and varies among strains. Leveraging these findings, we used comparative genomics and transcriptomics to identify candidate INA genes. Ten candidate INA genes that were predicted to encode secreted proteins were present only in the strains that produced the highest number of INPs. In total, 203 candidate INA genes coding for secreted proteins were induced at low temperatures. Among them, two genes predicted to encode hydrophobins stood out because hydrophobins are small, secreted proteins that form aggregates with amphipathic properties. We discuss the potential of the candidate genes to encode INA proteins and the next steps necessary to identify the molecular basis of INA in *F. avenaceum*.

## 1. Introduction

Homogeneous ice nucleation is the process by which pure water freezes in the absence of impurities, usually at temperatures below −38 °C [1]. In contrast, heterogeneous ice nucleation is the process by which ice formation occurs above −38 °C when particles with ice nucleation activity (INA), called ice-nucleating particles (INPs), serve as crystallization embryos. While the physical process of ice nucleation is poorly understood [2], it is well documented that INPs play an important role in atmospheric processes by affecting the ratio of frozen to liquid droplets in clouds, which in turn affects Earth’s radiation budget [3] and the formation of precipitation [4]. Some biological particles are particularly effective as INPs, inducing ice formation at temperatures ≥ −12 °C [5]; they have been found in clouds [6,7,8] and in precipitation [9,10,11]. While this suggests that biological particles may contribute to atmospheric processes, their relative contribution compared to inorganic particles, which are much less effective at ice nucleation but are present in clouds at much higher concentrations, is still unclear [12].

Biological INPs are associated with bacteria, fungi, viruses, pollen, lichen, and marine organics [5,13]. Bacterial INPs produced by strains in the Gram-negative bacterial genera of *Pseudomonas*, *Pantoea,* and *Xanthomonas* depend on homologs of the InaZ protein, which is associated with either the outer cell wall or with secreted vesicles [14,15,16,17,18,19,20,21,22,23,24]. InaZ and its homologs have been found to catalyze ice formation by affecting the order and dynamics of interfacial water and by removing latent heat [25,26]. The INA of the Gram-positive bacterium *Lysinibacillus parviboronicapiens* is instead associated with secreted INPs, the production of which depends on a type-I iterative polyketide synthase non-ribosomal peptide synthetase (PKS-NRPS) [27,28,29]. The mechanism by which *L. parviboronicapiens* INPs induce water to freeze is unknown. The question of how INA contributes to bacterial fitness is still under debate. In *Pseudomonas syringae*, it has been shown that strains with INA cause more frost damage [30] and lead to the more efficient ingress of bacteria from the leaf surface into intercellular spaces [31] than strains without INA. INA may also lead to a more efficient return of aerosolized bacteria from clouds back to the more nutrient-rich terrestrial environment [32].

Fungal INA was first found in the genus *Fusarium* [33] and later in *Mortierella*, *Puccinia*, and a small number of other genera [34,35]. *Fusarium* species are filamentous fungi, many of which are pathogens of plants and animals [36]. They are widely distributed in soils. but some are also dispersed over long distances through the atmosphere [37,38,39]. INA in the species *F. avenaceum* has been better characterized than in other species [33,39,40,41]. It is associated with secreted particles that consist of aggregates composed of subunits with a molecular weight lower than 100 kDa and a diameter as small as 6.1 nm [41]. *F. avenaceum* induces ice formation at temperatures as high as −1 °C [42], is stable at a pH ranging from 2 to 12, tolerates heat up to 40–60 °C, and is maintained under atmospherically relevant conditions and long-term storage [33,40,41]. *Fusarium* INPs have been proposed to be proteinaceous because of their heat sensitivity, peak UV absorbance at 280 nm [36], and loss of INA after proteinase treatment [40,43,44]. However, since polyketides and non-ribosomal peptides can also be sensitive to heat [45], UV absorbance at 280 nm is due to aromatic rings, which are also present in many polyketides [46]; *Fusarium* mycotoxins, which are polyketides, are enzymatically degradable [47,48], and so a PKS-NRPS could be at the basis of INA in *Fusarium* as well. The biological role of INA in fungi is still an open question [49].

A recent study found that approximately 16% of 112 *Fusarium* strains showed an INA above −12 °C. Strains with INA belonged to seven different *Fusarium* species, with most of these species also including strains without INA [41]. Therefore, it may be possible to identify candidate INA genes in *Fusarium* using a comparative genomics approach, based on their presence in strains that have INA and absence in strains without INA, as was performed to identify candidate INA genes in *L. parviboronicapiens* [27,28,29]. Additionally, if INA genes in *Fusarium* were to be induced under some conditions and suppressed under different ones, candidate INA genes could be identified by comparing gene expression between these conditions. Therefore, in this study, we screened 14 *F. avenaceum* strains for INA, further characterized the produced INPs and the conditions under which INA is induced, and performed a comparative genomics and transcriptomics study. Among the genes whose presence and expression correlated with INA, we focused on genes that encoded either secreted proteins or PKS-NRPS enzymes.

## 2. Materials and Methods

### 2.1. Fungal Strains

Twelve *F. avenaceum* strains were obtained from the Agricultural Research Service (ARS) culture collection; two strains were provided courtesy of David G. Schmale III (Virginia Tech), one strain of which was originally from Kansas State University (see Table 1 for details). All strains were grown on potato dextrose agar (PDA) prior to being processed.

### 2.2. INA Measurements

To obtain the cumulative ice nucleation spectra of each strain, 0.5 mg of mycelium was collected from the center of PDA plates and suspended in 1 mL of nuclease-free water. These primary suspensions were used to make dilution series in nuclease-free water from 10^−1^ to 10^−5^. Droplet-freezing assays were performed, using thirty drops of 20 µL volume of the primary suspension and of each dilution. Drops were deposited on Parafilm boats floating on a glycerol bath with a cooling thermostat (LAUDA Alpha Cooling Thermostat RA24, Lauda-Königshofen, Germany). The water used to make dilutions served as the negative control. INA was tested at −6 °C, −7 °C, −8 °C, −9 °C, −10 °C, −11 °C, and −12 °C. The drops were incubated for 10 min at each temperature, and the number of drops that froze in each group was recorded. The entire assay, starting from the preparation of suspensions, was repeated three times. The number of cumulative ice nuclei (IN) per gram of fungus at each temperature was inferred using the method developed by G. Vali [50] and described by K. C. Failor et al. [27]. An analysis of variance (ANOVA) was performed using R (v4.0.4) to determine if there were statistically significant differences in INA among strains and treatments.

### 2.3. Characterization of INPs

The *F. avenaceum* strain F156N33 was used to investigate the properties of INPs. After the strain was grown on PDA for 7 days at room temperature, a primary suspension was made by suspending 5 mg of mycelium from the center of each plate in 50 mL of nuclease-free water. Next, 49 mL of the primary suspension was passed through a 0.22-μm-pore-size filter (Millex-GP Syringe Filter, 0.22 µm, Darmstadt, Germany) to obtain the 0.22 μm filtrate. Then, 20 mL of the 0.22 μm filtrate was passed through a 30-kDa-pore-size filter (Macrosep Advance Centrifugal Devices with Omega Membrane 30K, Port Washington, NY, USA) for 10 min at 5000 rpm. This step separated INPs below approximately 5 nm in size, since these INPs could pass through the 30 kDa filter and end up in the filtrate. INPs above approximately 5 nm in diameter were retained by the 30 kDa filter and could be resuspended from the filter; these constituted the 30 kDa retentate. The retentate was resuspended from the filter membrane with 500 μL of nuclease-free water. In parallel, another 20 mL of the 0.22 μm filtrate was passed through another 30 kDa filter with the same centrifugation setting; however, 10 mL of nuclease-free water was added to the filter and the filters were centrifuged again. This washing step was performed a total of ten times. The final 30 kDa filter retentate was obtained by resuspending the retentate in 500 μL of nuclease-free water. The primary suspension, the 0.22 μm filtrate, the 30 kDa filtrate, the original 30 kDa retentate, the final 30 kDa retentate, the tenth filtrate, and 10^−1^ to 10^−5^ dilutions of each fraction were used to infer the cumulative ice nucleation spectra.

### 2.4. Investigating the Effect of Growth Conditions on INA

The *F. avenaceum* strain F156N33 was also studied to investigate the effect of growth temperature. The strain was grown for approximately 30 days at 6 °C, at room temperature, or at 28 °C, respectively. To establish the effect of culture age, the strain was grown for 7 days, 14 days, 21 days, 28 days, and 35 days, respectively, always at room temperature. The primary suspension and 10:1 to 10:5 dilutions were made as described above for each of the treatments and were used to infer the cumulative ice nucleation spectra, as described above.

### 2.5. Genome and Transcriptome Sequencing and Assembly

The genomic DNA of all 14 *F. avenaceum* strains was extracted from mycelium grown on PDA using the ZymoBIOMICS DNA Miniprep Kit (Zymo Research, Irvine, CA, USA). The total RNA of the *F. avenaceum* strain F156N33 was extracted using the RNeasy^®^ Plant Mini Kit (QIAGEN, Hilden, Germany) after INA was confirmed, to ensure that the INA genes were expressed. DNA and RNA were sequenced on an Illumina Nova Seq 6000 Platform at the Novogene Corporation Inc. (Sacramento, CA, USA). Low-quality reads and adapters were removed by the company. The quality of reads was checked using FastQC v0.11.9 [51]. The methods used for genome and transcriptome assembly have been described in detail in S. Yang et al. [52].

### 2.6. Phylogenetic Analyses

The assemblies of 14 *F. avenaceum* strains were used, and the *F. tricinctum* strain INRA 104 (GenBank accession: OVTS00000000) served as the outgroup. Sequences of translation elongation factor 1-alpha (*TEF-1α*), the RNA polymerase II largest subunit (*RPB1*), and the RNA polymerase II second-largest subunit (*RPB2*) were identified by BLASTN v2.10.0+ [53] using a custom database containing sequences of these genes in other *Fusarium* species obtained from GenBank (*TEF-1α*: FFUJ_05795 from *F. fujikuroi*; *RPB1*: FFUJ_00736 from *F. fujikuroi*; *RPB2*: FFUJ_07996 from *F. fujikuroi*). Multiple sequence alignments for each of the three genes were performed using Clustal W [54] in MEGA 7 [55] with the default setting, and the resulting alignments were manually edited.

Phylogenetic analyses were performed using the maximum likelihood method in MEGA 7 [55] for each of the three genes, as well as a concatenated gene dataset. The best nucleotide substitution model was determined for each of the single genes and the combined dataset using MEGA 7 [55]. The maximum likelihood trees for each of the single genes and the combined dataset were generated using the best nucleotide substitution model, accordingly, with 1000 bootstrap replications in MEGA 7 [55].

### 2.7. Gene Prediction and Genome Annotation

The assembled 14 genomes were annotated using the MAKER annotation pipeline (v3.01.03) [56] with a combination of evidence-based methods and ab initio gene prediction, as previously described [52]. In brief, for each genome assembly, the previously assembled transcriptome of F156N33 by both Trinity and StringTie served as EST evidence, while the proteome of *Fusarium graminearum* (UniProt Proteomes accession: UP000070720) served as protein homology evidence. Ab initio gene annotations were performed by SNAP v2013-02-16 [57] and AUGUSTUS v3.4.0 [58] afterward.

### 2.8. Functional Annotation, Prediction of the Signal Peptide and Prediction of Secondary Metabolite Genes

Functional annotation was performed using InterProScan v5.46-81.0 for the presence of Pfam domains, with terms from the Gene Ontology [59]. BLASTP from BLAST v2.10.0+ [53] was also used to find regions of local similarity against the February 2021 release of the Swiss-Prot database [60]. Prediction of the signal peptide was performed using SignalP v5.0b [61]. Phyre2 was used to predict protein function based on the homology of predicted protein structures with the structure of proteins with known function [62]. Prediction of metabolic gene clusters was performed by the fungal version of antiSMASH 6 [63].

### 2.9. Pan-Genome Analysis and Prediction of Orthologs

Pan-genome analysis and the search for orthologous genes in the genome of 14 *F. avenaceum* strains were performed by GET_HOMOLOGUES-EST v3.4.2 [64], which clusters homologous gene families using the OrthoMCL v1.4 [65] clustering algorithm. Genes present in strains with INA and strains without INA were identified using the script (parse_pangenome_matrix.pl) included in this program.

### 2.10. Gene Expression Analysis

As described above, RNA-seq reads from the *F. avenaceum* strain F156N33 grown at 6 °C (3 replicates) and at room temperature (3 replicates) were obtained. Each replicate was aligned to the genome assembly of F156N33 using STAR v2.7.8a [66], generating five alignment files in BAM format. These five BAM files were subjected to featureCounts v2.0.1 [67] with the parameters -p -B -C (multi-mapped reads excluded), as well as -p -B -O -M (multi-mapped reads included) to determine the number of reads mapped to each gene. The read counts were normalized by the DESeq2 package in R [68]. Thus, differentially expressed genes (DEGs) were identified with the following parameters: “padj (adjusted *p* value) < 0.05 and log2FoldChange > 1 using DESeq2.”

### 2.11. DNAseq Analysis

The *F. avenaceum* strain F156N33 served as the reference genome for DNAseq, which allowed us to determine the presence and absence of each gene of strain F156N33 in each of the 13 other genomes, based on read alignment, to independently confirm the GET_HOMOLOGUES-EST v3.4.2 [64] results. DNA reads were mapped on the reference genome using BWA-MEM2 v2.2.1 [69]. The mapping quality was assessed by Qualimap v2.2.2 [70] and the number of reads mapped to each gene was determined by featureCounts v2.0.1 [67] with parameters -p -B -O -M. The alignment files were processed by the Genome Analysis Toolkit v4.0 (GATK) [71]. GATK “SortSam” and “MarkDuplicates (Picard)” were used to remove the duplicated mapping reads.

## 3. Results and Discussion

### 3.1. INA Varies between Strains of F. avenaceum

To determine the distribution of INA within the species *F. avenaceum*, cumulative ice nucleation spectra were obtained for 14 strains, as shown in Table 1. All strains presented INA to some extent. Thirteen strains started to freeze around −6 °C to −7 °C, while the freezing of *F. avenaceum* NRRL 54396 was only observed at −9 °C and below (Figure 1). Among the 13 strains that started to induce freezing at −6 °C to −7 °C, two of them (NRRL 13826 and NRRL 36457) showed significantly lower activity than the other 11 strains, which started freezing at −6 °C to −7 °C (*p*-value = 3.73 × 10^−5^ at −6 °C). However, the 11 most active strains still varied significantly, based on the inferred cumulative number of IN per gram of mycelium, with 4 strains (NRRL 66272, NRRL 13316, NRRL 54754, and F156N33) having very similar and higher INA at the lowest tested temperatures, compared to the other 7 strains. With regard to the cumulative number of IN produced per gram of mycelium, the 11 most active strains produced 10^7^ to 10^10^  IN/g at −8 °C and below, with *F. avenaceum* NRRL 66272 showing the absolute highest number of IN/g.

Because INA was detected in all strains, with the number of IN/g of mycelium varying gradually between strains, we conclude that INA in *F. avenaceum* is rather a quantitative trait than a qualitative trait. The strength of INA may depend on the presence or absence of several INA genes and/or be affected by allelic differences in one or several INA genes. Additionally, neither the substrate of isolation (plants, soil, or atmosphere) nor geographic location appears to be correlated with INA.

### 3.2. F. avenaceum INPs Are Secreted Aggregates Prone to Separation by Centrifugation and Washing

To investigate the properties of *F. avenaceum* INPs, filtration was performed for the primary suspension of strain F156N33 using filters with two different pore sizes (0.22 μm and 5 nm, the approximate pore size of a 30 kDa filter). The 0.22 μm filtrate showed a high INA with approximately 10^6^ IN per gram of mycelium at −7 °C and reaching 10^8^ IN/g at −12 °C. This is only a tenfold reduction compared with the primary mycelial suspension of strain F156N33.

After passing through a 30 kDa filter, the number of IN/g was reduced approximately 10,000 to 100,000-fold at −7 °C (*p*-value = 0.0381) and −8 °C (*p*-value = 0.0392) and still 1000-fold at −9 °C to −12 °C (for example, *p*-value = 0.0066 at −9 °C) compared with the primary suspension (Figure 2). The retentate collected from the 30 kDa filter was reduced tenfold at −7 °C (*p*-value = 0.0152) and −8 °C (*p*-value = 0.0921), compared with the 0.22 μm filtrate, but presented an INA similar to the 0.22 μm filtrate at lower temperatures (for example, *p*-value = 0.3500 at −9 °C). This suggests that *F. avenaceum* INPs are secreted from mycelial cells and that most *F. avenaceum* INPs have a diameter of at least 5 nm, corresponding to a mass above 30 kDa.

Intriguingly, the combined 30 kDa retentate and the 30 kDa filtrate had a higher INA than that of the primary suspension, in particular at −12 °C, at which temperature we found 10^14^  IN/g. This suggests that during filtration, larger *Fusarium* INPs may have separated into smaller INPs. To follow up on this hypothesis, we washed the 30 kDa filter 10 times. In other words, after the first centrifugation, water was added to the filter, and centrifugation was repeated. This process was then performed another 9 times. Each filtrate was tested for INA and the final retentate was tested for INA as well. Even after 10 washes, the cumulative number of IN per gram of original mycelium for the 30 kDa retentate was still 10^7^/g at −12 °C, only approximately 10-fold lower compared to the first, unwashed 30 kDa retentate (*p*-value = 0.2410) (Figure 2). Moreover, the filtrate from the tenth wash still showed similar INA to the final, washed 30 kDa retentate (*p*-value = 0.0835). This suggests that INPs even smaller than 5 nm (~30 kDa) are generated from the original INPs during each washing and centrifugation step. Therefore, INPs secreted by *F. avenaceum* appear to consist of aggregates made of individual units smaller than ~30 kDa. This is consistent with the results reported by A. T. Kunert et al. [41], who performed filtration experiments with a 100 kDa filter and concluded that *Fusarium* INPs consist of aggregates of individual macromolecules smaller than 100k Da.

### 3.3. Growth Temperature Affects Fusarium INA, while the Length of Growth Time Does Not Have a Significant Impact

To investigate how growth temperature affects INA in *F. avenaceum*, strain F156N33 was grown at different temperatures for about 30 days, prior to determining the cumulative IN spectra. Observing the mycelia by eye, it could be seen that growth temperature affected their morphology (Appendix A). Although growth was slow at 6 °C, the mycelium covered the entire plates after 30 days of growth at 6 °C. When grown at 28 °C instead, the mycelium never covered more than half the plate.

In terms of INA, INA was higher when grown at 6 °C compared to growth at room temperature, while it was reduced after growth at 28 °C (Figure 3A). More precisely, the cumulative number of IN/g of mycelium at −12 °C was around 10^13^ when grown at 6 °C, at 10^9^ when grown at room temperature, and at only 10^6^ when grown at 28 °C (*p*-value = 0.0015).

Previous studies on bacterial INA also reported that low temperature (15 °C) induced INA in *Pseudomonas syringae*, although an even lower temperature (9 °C) inhibited INA. How temperature induces bacterial INA was not shown conclusively [72,73]. We hypothesize that INA in *F. avenaceum* may be higher at lower temperatures because the expression of INA genes may be induced at lower temperatures and repressed at higher temperatures. The higher expression of INA genes may then lead to a higher production of INPs. On the other hand, it is also possible that the structure or size of INPs that forms at lower growth temperatures is different from the structure or size of INPs formed at higher temperatures. Finally, post-translational factors could be involved that alter the surface of INPs at lower growth temperatures.

The impact of culture age was also investigated using *F. avenaceum* F156N33 grown for 35 days at room temperature. While no significant change in INA was observed between the 7-day, 14-day, and 21-day time points, the cumulative IN/g dropped approximately tenfold after 28 and 35 days of growth, in particular at temperatures from −8 °C to −12 °C, although not significantly (for example, *p*-value = 0.3190 at −8 °C) (Figure 3B). At −6 °C, INA was inconsistent among replicates, making it challenging to compare INA between cultures grown for different lengths of time. This result suggests that *F. avenaceum* continues to produce INPs as long as the mycelium grows.

### 3.4. Phylogenetic Analyses Reveal That F. avenaceum Strains Form Several Within-Species Clusters and That the Strength of INA Does Not Correlate with Phylogeny

Whole-genome sequencing and genome assembly were performed for all 14 *F. avenaceum* strains. The genome coverage of the assemblies ranged from 49× to 61×. Assembly sizes ranged from 36.8 Mb to 49.7 Mb, and the G+C content ranged from 48.20% to 48.50%, with one exception (50.72% for strain NRRL 54396). The BUSCO quality assessment was based on the lineage-specific profile library 9ypocreales_odb10 (4494 genes) and revealed that more than 97.6% of genes were present in all 14 assemblies, indicating a high quality of genome assembly for all strains. The assembly statistics are shown in Table 2.

All 14 strains were identified as *F. avenaceum*, based on BLASTN. The sequences of translation elongation factor 1-alpha (*TEF-1α*), RNA polymerase II largest subunit (*RPB1*), and RNA polymerase II second-largest subunit (*RPB2*) were extracted from each of the 14 assemblies to perform phylogenetic analyses. Since *F. tricinctum* is a closely related species, the genome of reference strain *F. tricinctum*, INRA104, was chosen as the outgroup.

All *F. avenaceum* strains formed one large clade, separated from the outgroup *F. tricinctum*. Three to four major sub-clades formed, depending on the genes that were used (Figure 4). The *TEF-1α* gene sequences had the least genetic diversity and the lowest phylogenetic resolution, with six strains having an identical sequence and forming a single clade with one additional strain (Figure 4A), while the *RPB1*-based ML tree and the multilocus sequence tree had the highest genetic diversity (Figure 4B,C).

When comparing the trees with the geographic origin of strains and their substrates of isolation (see Table 1 and Figure 4), the strains did not cluster together based on where and from what they were isolated. In addition, although the three strains, NRRL 13826, NRRL 36457, and NRRL 54396, with the relatively lowest INA were phylogenetically distinct from our reference strain F156N33; according to all four ML trees, they clustered together with other strains with high INA, for example, strain NRRL 66272.

In summary, based on the four genes used for phylogenetic analysis, the strength of INA in *F. avenaceum* does not correlate with phylogeny. This could either be due to convergent evolution, with multiple independent gene acquisitions (of the same or different INA genes) or independent mutations (in the same or different INA genes) that increase INA or to multiple gene loss events or multiple mutations that lead to a decrease in INA.

### 3.5. A Comparative Genomics Approach to Identifying Candidate INA Genes in F. avenaceum

*F. avenaceum* F156N33 was annotated using the collected RNA-seq data. Thus, it served as the reference genome in our study, and 11,233 genes were predicted in F156N33. Since we hypothesized that *F. avenaceum* INPs are either secreted protein aggregates or consist of the products of PKS–NRPS gene clusters, a list of putative INA genes was obtained: 1155 genes were predicted to encode proteins with signal peptides and 59 genes were predicted to belong to PKS-NRPS gene clusters. Therefore, a total of 1214 genes are candidate INA genes in F156N33 (Appendix A).

To reduce the number of candidate INA genes in *F. avenaceum*, we performed a pan-genome analysis. To do this, orthologous groups were identified for all 14 strains and the genes were clustered into orthologous groups. One predicted transcript (Gene ID: KAF25_002500) was skipped by the program because it was longer than 25,000 bp. In total, 554 genes were identified as redundant isoforms. Among 10,678 ortholog groups, 8210 orthologs were found to be present in all 14 strains. We then compared the presence and absence of genes with the strength of INA to test the general hypothesis that one or more INA genes would only be present in the strains with the highest INA. Gene presence/absence results were confirmed with a read-based DNAseq analysis (Appendix A).

We first hypothesized that one or more INA genes may be present in strain F156N33 and the other 3 most active strains (NRRL 13316, NRRL 54754, and NRRL 66272) and absent from the least active strain, NRRL 54396, ignoring all strains with intermediate INA. In total, 82 genes were thus identified (Appendix A).

Based on the phenotypic results, *Fusarium* INPs were likely to be secreted molecules. Therefore, we assumed that *Fusarium* INPs were secreted proteins. Ten genes among these 82 genes were predicted to encode proteins with signal peptides. Four of these genes had no annotation or were annotated as proteins of unknown function, based on InterProScan or BLASTP. Five genes were found to encode enzymes, and most of them belonged to glycosyl hydrolase families. One gene was found to be a hydrophobic surface, binding protein A, annotated as cell wall mannoprotein 1 by BLASTP. According to Phyre2, the four unknown proteins were predicted to encode enzymes, two of which were hydrolases (Appendix A). One gene was found to encode a homolog of the Aspergillus hydrophobic surface binding protein A (HsbA) [74], annotated as cell wall mannoprotein 1 by BLASTP. Since we do not expect hydrolases to have INA, this gene was the most promising INA gene among the 10 genes predicted to encode secreted proteins. Moreover, the molecular weight of HsbA (14 kDa) fits our phenotypic characterization of *F. avenaceum* INPs, and its homolog in *Aspergillus* was confirmed to be secreted [74].

Since polyketide non-ribosomal peptides are a kind of biological INP [29] and PKS-NRPS gene clusters are involved in the biosynthesis of mycotoxins in *Fusarium* [75], we assumed that PKS-NRPS gene clusters could also be involved in producing INPs in *Fusarium*. One gene among the 82 candidate genes was predicted to be a PKS-NRPS gene, and another three genes fell into one PKS-NRPS cluster. The PKS-NRPS gene (Gene ID: KAF25_005439) was predicted to encode an alcohol dehydrogenase. The cluster this gene belonged to was most similar to the fusaridione A biosynthetic gene cluster from *Fusarium heterosporum*. The cluster that the other three genes belonged to was most similar to the bikaverin biosynthetic gene cluster from *Fusarium fujikuroi* IMI 58289. While we cannot exclude the possibility that these genes contribute to INA, their presence in *Fusarium* species without INA does not make them strong candidates.

Next, we tested a slightly different hypothesis. We assumed again that the strain with the lowest INA, strain NRRL 54396, was truly INA-negative and had no INA gene, but that INA genes were present in all 11 strains with either high or medium activity. Only 23 genes were so identified (Appendix A), and they were a subset of the above 82 genes with one of them being the gene encoding the HsbA homolog in Appendix A. Only two genes were predicted to encode proteins with signal peptides; one (Gene ID: KAF25_008711) was a protein of unknown function by InterProScan or BLASTP and was predicted to encode a hydrolase (cellulose-binding protein) by Phyre2. Another gene (Gene ID: gene-KAF25_007828) encoded the hydrophobic surface-binding protein A, mentioned above.

Finally, we tested the hypothesis that some INA genes may be present in all but the three strains with the lowest INA. Only one gene (Gene ID: KAF25_004243) met this criterion, based on the pan-genome analysis. However, the probability of this gene encoding a secreted peptide was only 0.0854% and it was not predicted to be a biosynthetic gene. Therefore, this gene is unlikely to be a candidate INA gene.

In conclusion, most genes that are present in the most active strains but absent from the least active strain(s) encoded enzymes with a predicted function, mostly hydrolases. We think that such enzymes are unlikely to form aggregates that have INA. This leaves the HsbA homolog as the main candidate INA protein. However, in the absence of experimental data, none of the identified genes should be excluded at this time. Besides, one or more of the genes encoding proteins that have signal peptides but are of unknown function remain promising INA gene candidates. It will be interesting to predict their structure and determine if any of them may possess characteristics that allow them to form aggregates and/or that have other characteristics that are predicted to make surfaces ice nucleation-active [76].

It is also possible that one or more of the 8210 genes present in all 14 strains may be responsible for INA and that the difference in INA is due to allelic differences in some of these genes. Or, as we will investigate next, INA genes may be present in all strains but be expressed at a significantly higher level at low temperatures, when INA was found to be highest, compared to higher temperatures, when INA was found to be lower. However, to find statistically significant associations between gene (alleles) and INA, many more strains would need to be used in a genome-wide association study (GWAS) [77].

### 3.6. Transcriptomics Approach to Identify Candidate INA Genes in F. avenaceum

Based on our phenotyping results, the INA in *F. avenaceum* F156N33 was higher after growth at 6 °C than growth at room temperature or at 28 °C. Therefore, we hypothesized that among the putative genes in strain F156N33, the genes that were more highly expressed at 6 °C than at room temperature, were more likely to be INA genes than the genes that were expressed higher at room temperature rather than at 6 °C, or the genes that were equally expressed at both temperatures. We could not include gene expression data at 28 °C since the slow growth of *F. avenaceum* at 28 °C did not allow us to extract enough RNA for sequencing.

Differential expression (DE) analyses were performed for uniquely mapped reads only, as well as by including multi-mapped reads. Multi-mapped reads did not make much of a difference. In fact, 984 genes were found upregulated at 6 °C for uniquely mapped reads only, and 1032 genes were found when multi-mapped reads were included. In total, 1037 genes were upregulated at 6 °C (Appendix A).

Similarly to the comparative genomics analysis above, we looked next at which of the 1037 differentially regulated genes were either predicted to encode proteins that contained signal peptides or that fell within biosynthetic gene clusters.

In total, 203 genes among the 1037 genes were predicted to encode proteins with signal peptides (Appendix A). Of these genes, 65 had no annotation or are uncharacterized, based on InterProScan or BLASTP. In total, 109 genes were predicted to encode enzymes. Many of them were again predicted to be hydrolases. Interestingly, a putative INA gene of *F. acuminatum* was reported in previous studies to encode a protein with INA when expressed in *E. coli* [78,79] and belongs to the glycoside hydrolase family 16, based on InterProScan. However, this gene was originally identified using the *P. syringae* INA gene as a hybridization probe, although the gene has no actual homology to the *P. syringae* INA gene. Moreover, the reported INA test results for this protein did not include a negative control, a cumulative IN spectrum was not reported, nor were results repeated independently. Therefore, it is unlikely that this protein has INA.

The remaining 29 genes included genes encoding site-specific binding proteins, genes encoding specific domains, virulence factors, and genes involved in the production of molecules, such as antifungal proteins, mycotoxins, and fungal hydrophobins.

The two genes coding for secreted hydrophobins caught our attention (Figure 5). The gene KAF25_008764 was significantly higher in expression at 6 °C for both the uniquely mapped reads and multi-mapped reads. It has a predicted molecular weight of 14.4 kDa and was annotated by BLASTP as a rodlet protein; Phyre2 found this protein to align with high confidence over most of its length with the following class I hydrophobins: the Roda hydrophobin protein of *Aspergillus fumigatus*, the Hyd1 hydrophobin from *Schizophyllum commune*, and the hydrophobin Mpg1 from the rice blast fungus, *Magnaporthe oryzae*. Hydrophobins self-assemble into amyloid-like fibrillar aggregates called rodlets [80], which form monolayers at hydrophobic:hydrophilic interfaces. For example, some rodlets coat the spore surfaces, turning them from hydrophilic to hydrophobic, thereby keeping spores dry and facilitating dispersal under wet conditions [81]. We hypothesize that the hydrophobin molecules secreted by *F. avenaceum* could form small rodlets that self-assemble into larger aggregates (or that coat particles made of other molecules). Because of their strong amphipathic properties, they could then induce changes to the structure of liquid water and facilitate crystallization, similar to what was found for the InaZ protein [25,26]. Since we found that *F. avenaceum* INPs can pass through the 30 kDa filter and that individual KAF25_008764 proteins already have a predicted molecular weight of 14.4 kDa, it may also be possible that a single KAF25_008764 protein is the smallest INP, which can reversibly aggregate into larger particles. However, typical class-I hydrophobins can only depolymerize after treatment with strong acids [82]. While the KAF25_008764 gene is present in all *F. avenaceum* strains, phylogenetic analysis revealed that two strains with the relatively lowest INA (NRRL 13826 and NRRL 54396) were clustered together away from other more active strains (Figure 6A), suggesting that allelic differences in this gene could at least partly explain the differences in INA between strains.

The second gene coding for a predicted hydrophobin, KAF25_005037, was significantly higher in expression at 6 °C for multi-mapped reads only, has a molecular weight of 10 kDa, and is predicted to be a class-II hydrophobin, which does not require strong acids to depolymerize. However, the allelic sequences of this gene, which is also present in all 14 genomes, did not correlate with the strength of INA (Figure 6B).

With regard to the 65 genes of unknown function, according to Phyre2, 59 had predictions (Appendix A). In total, 16 genes were predicted to encode enzymes, 10 of which encoded hydrolases. The remaining 43 genes found here were involved in cell transport, transcription, binding, or the production of certain proteins, such as mycotoxins, hormones, and antimicrobial proteins. However, some predictions had very low confidence, so the functions of these genes are uncertain.

As for genes in PKS-NRPS clusters, 48 genes among the 1037 genes upregulated at 6 °C were predicted to fall into 23 clusters (Appendix A). Of these 48 genes, seven were predicted to be PKS-NRPS genes, distributed among seven clusters. Most of these seven genes were predicted to encode AMP-binding enzymes. Four genes (Gene IDs: KAF25_000205, KAF25_001891, KAF25_002500, KAF25_011137) were associated with polyketide synthases or polyketide synthase dehydratases.

Finally, we compared the results from the comparative genomics analysis with the results from the transcriptomics analysis to check for any overlap. Only one gene encoding for a protein with a signal peptide was so identified, and then, only if we assume that INA genes are absent from the least active strain and are present in the four strains with the highest INA. This gene (KAF25_006087) is predicted to encode a peptidase/proteinase and is, thus, an unlikely INA gene candidate.

In summary, many genes upregulated at 6 °C in *F. avenaceum* are predicted to encode metabolic enzymes. Therefore, they are unlikely candidate INA genes. However, other genes were found to be either part of biosynthetic gene clusters predicted to produce polyketide non-ribosomal peptides, encode secreted proteins of as yet unknown function, or, in one case, encode a predicted secreted hydrophobin, a class of proteins known to self-assemble into larger aggregates. While the hydrophobins fit our phenotypic results the best, in terms of the comparative genomics results, it would be premature to exclude any of the other genes from being putative INA genes since most of them are poorly characterized at this point.

## 4. Conclusions

In this study, we found that although all available *F. avenaceum* strains could secrete INPs, the ability to induce ice formation varied among strains. This suggests that either the presence or absence of several genes contributes to the strength of INA in *F. avenaceum*, or that allelic differences in one or more INA genes affect the INA. Moreover, INA in *F. avenaceum* is associated with secreted aggregates that appear to consist of subunits as small as 5 nm in diameter. INA in *F. avenaceum* is higher at lower temperatures, suggesting that the expression of INA genes may be induced at lower temperatures and may be repressed at higher temperatures. Comparing the gene contents of strains with different strengths of INA and the expression of genes at different temperatures, we have obtained a list of putative INA genes that either encode the secreted proteins or are located in biosynthetic gene clusters. However, genes outside of this list cannot be ruled out because allelic differences in the genes present in all strains may be the main determinants of INA in *F. avenaceum*. Unfortunately, fourteen strains are not sufficient to test this hypothesis by performing a genome-wide association study (GWAS) [77], and we do not have access to any additional strains at this point. Therefore, to ultimately identify the INA genes in *F. avenaceum*, either more strains will need to be analyzed to conduct GWAS and/or candidate INA genes need to be confirmed experimentally through either mutational analysis in *F. avenaceum* or via gain-of-function experiments, expressing them in *Fusarium* species that do not have INA.

At this point, the genes predicted to encode hydrophobins that are upregulated at 6 °C are the most promising candidate INA genes. They have the anticipated size and are predicted to be secreted, form aggregates, and have characteristics consistent with a molecule that can affect the structure of liquid water. These genes and other candidate INA genes herein identified will need to be investigated experimentally through either mutational analysis in *F. avenaceum,* followed by screening for the loss of INA, or gain-of-function experiments expressing them in *Fusarium* species without INA and screening for the gain of INA. Once the INA genes in *F. avenaceum* have been identified, INA genes in other *Fusarium* species and possibly in other fungal genera may be identified by homology, and the molecular structure and composition of fungal INPs could be more easily determined. Surveying clouds for the presence of fungal INA genes and/or the INPs they produce could then help to unravel the role of fungal INA in atmospheric processes. Finally, identifying the *Fusarium* INA genes and their products could lead to an improved understanding of the process of INA itself, which is still poorly characterized [49].

## Figures and Tables

**Figure 1 jof-08-00958-f001:**
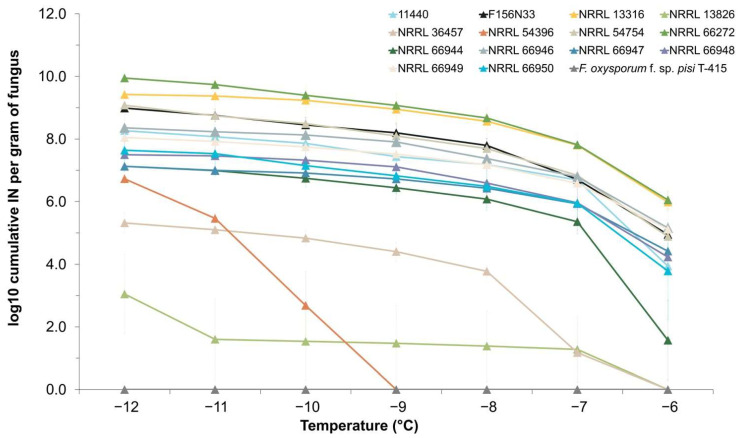
Cumulative ice nucleation spectra of 14 *Fusarium avenaceum* strains. All cultures were grown at room temperature for 7 days. The results are of primary suspensions based on droplet-freezing assays at −6, −7, −8, −9, −10, −11, and −12 °C. Each data point represents a mean number (±SEM) obtained from three replicates. IN: ice nuclei.

**Figure 2 jof-08-00958-f002:**
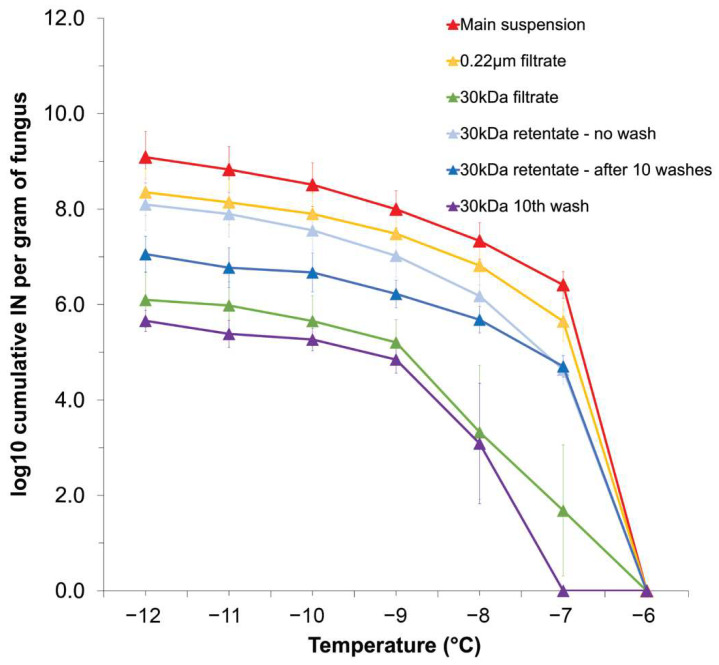
Cumulative ice nucleation spectra of *F.*
*avenaceum* F156N33, grown at room temperature for 7 days. The results are of primary suspensions, 0.22 μm filtrates, 30 kDa filtrates, original 30 kDa retentates, washed 30 kDa retentates, and last washes, based on droplet freezing assays. Each data point represents a mean number (± SEM) obtained from three replicates. IN: ice nuclei.

**Figure 3 jof-08-00958-f003:**
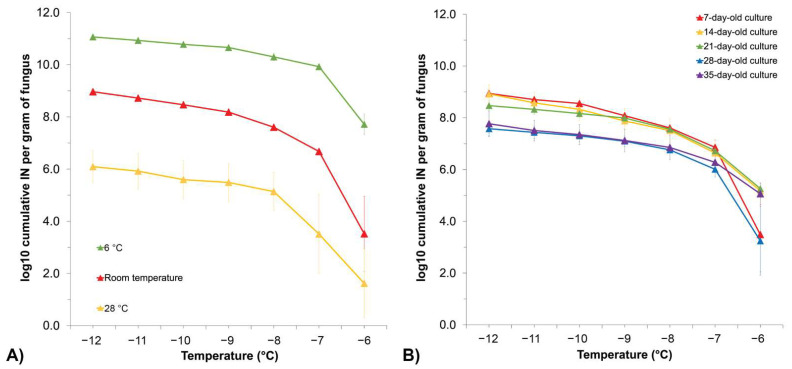
Cumulative ice nucleation spectra of *F.*
*avenaceum* F156N33 (**A**) grown at room 6 °C, temperature, and 28 °C, respectively, for about 30 days; (**B**) grown at room temperature for 7 days, 14 days, 21 days, 28 days, and 35 days, respectively. The results are from primary suspensions based on droplet freezing assays. Each data point represents a mean number (±SEM) obtained from three replicates. IN: ice nuclei.

**Figure 4 jof-08-00958-f004:**
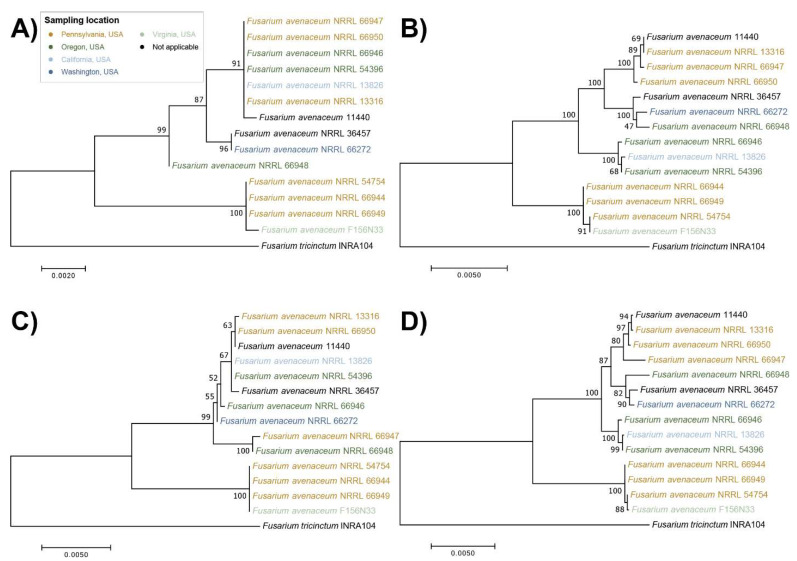
Maximum likelihood (ML) trees, constructed based on the sequences of (**A**) translation elongation factor 1-alpha (*TEF-1α*), (**B**) RNA polymerase II largest subunit (*RPB1*), (**C**) RNA polymerase II second-largest subunit (*RPB2*), and (**D**) combined four-locus data set, using the best nucleotide substitution model with 1000 bootstrap replications. Each color represents a state where the strain was isolated.

**Figure 5 jof-08-00958-f005:**
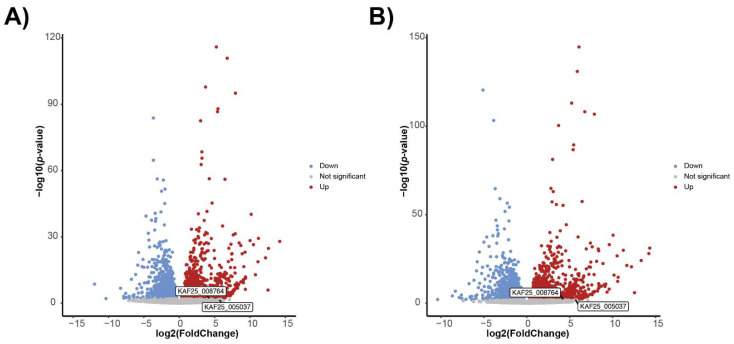
Volcano plots for genes upregulated at 6 °C (**A**) using uniquely mapped reads and (**B**) using multi-mapped reads. Labels indicate two secreted hydrophobins.

**Figure 6 jof-08-00958-f006:**
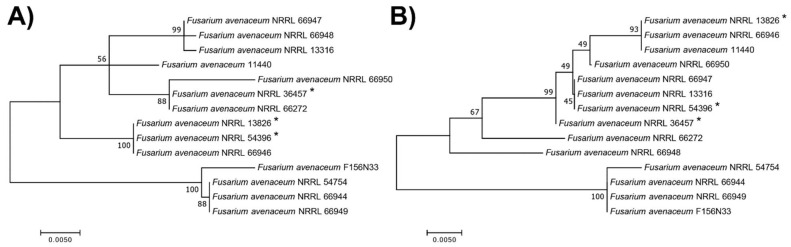
Maximum likelihood (ML) trees, constructed based on the sequences of two genes coding for secreted hydrophobins, (**A**) KAF25_008764 and (**B**) gene KAF25_005037, using the best nucleotide substitution model with 1000 bootstrap replications. * indicates the strains with the lowest INA.

**Table 1 jof-08-00958-t001:** List of *F. avenaceum* strains tested for INA.

Source	Accession	Substrate	Sampling Location	Accession(s) in Other Collections
USDA ARS Culture Collection (NRRL)	13316	Turf soil	Pennsylvania, USA	NA
	13826	Carnation	California, USA	NA
	36457	Barley kernel	USA	CBS 409.86/FRC R-8509/IMI 309353
	54396	Soil	Easter Lilly Research Borrkings, Oregon, USA	F49
	54754	Corn	Pennsylvania, USA	A-28077
	66272	Wheat	Washington, USA	A-28073
	66944	Seedling of spruce	Pennsylvania, USA	A-28042
	66946	Plant roots, Douglas fir tree	Oregon, USA	A-28020
	66947	Seedling of Douglas fir	Pennsylvania, USA	A-28035
	66948	Sugar pine tree seedling	Oregon, USA	A-28040
	66949	Seedling of spruce	Pennsylvania, USA	A-28041
	66950	Seedling of spruce	Pennsylvania, USA	A-28043
Kansas State University	11440	NA	NA	
Virginia Tech	F156N33	Atmosphere	Virginia, USA	

**Table 2 jof-08-00958-t002:** Assembly summary of 14 *F. avenaceum* strains.

Strain	Coverage (×)	Assembly Size (bp)	Number of Contigs	Maximum Contig Length (bp)	Minimum Contig Length (bp)	Average Contig Length (bp)	Median Contig Length (bp)	N50 Contig Length (bp)	GC Content (%)	^a^ Assembly BUSCO Coverage (%)
F156N33	51	41,175,306	214	3,233,628	210	192,487	1075	1,472,944	48.44	C:97.8; F:0.5; M:1.7
11440	52	42,933,485	897	1,984,632	200	48,011	538	1,024,532	48.33	C:97.7; F:0.5; M:1.8
NRRL 13316	61	41,704,585	964	2,164,424	200	43,399	579	843,661	48.26	C:97.7; F:0.5; M:1.8
NRRL 13826	51	44,694,304	1465	1,780,814	200	30,660	493	779,627	48.35	C:97.6; F:0.5; M:1.9
NRRL 36457	54	38,761,238	308	2,047,741	234	125,928	1068	918,031	48.38	C:97.7; F:0.6; M:1.7
NRRL 54396	51	49,692,405	626	1,709,761	200	79,506	659	691,401	50.72	C:97.8; F:0.5; M:1.7
NRRL 54754	50	38,889,550	234	4,244,225	229	166,303	875	1,329,635	48.48	C:97.8; F:0.5; M:1.7
NRRL 66272	49	39,140,173	251	2,646,538	226	156,028	1000	1,190,042	48.47	C:97.6; F:0.5; M:1.9
NRRL 66944	59	36,826,384	216	3,227,342	231	170,590	845	1,246,826	48.48	C:97.6; F:0.4; M:2.0
NRRL 66946	54	40,603,425	397	2,016,945	200	102,387	945	972,920	48.42	C:97.7; F:0.5; M:1.8
NRRL 66947	61	40,212,181	320	2,482,218	228	125,771	996	1,127,118	48.34	C:97.9; F:0.5; M:1.6
NRRL 66948	57	38,139,861	290	2,379,904	202	131,584	1103	1,046,193	48.31	C:97.9; F:0.5; M:1.6
NRRL 66949	52	37,127,996	260	3,604,089	234	142,881	1021	1,215,522	48.41	C:97.7; F:0.4; M:1.9
NRRL 66950	51	40,854,987	317	4,247,833	214	128,969	1117	1,141,783	48.38	C:97.8; F:0.5; M:1.7

^a^ For assembly BUSCO coverage, C stands for complete BUSCOs, F stands for fragmented BUSCOs, and M stands for missing BUSCOs.

## Data Availability

All original sequencing data were submitted to NCBI with accession numbers PRJNA720629 and PRJNA849802.

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
