# Peer review of "Identification of Candidate Ice Nucleation Activity (INA) Genes in *Fusarium avenaceum* by Combining Phenotypic Characterization with Comparative Genomics and Transcriptomics"

_jof, 2022, doi:10.3390/jof8090958_

Round 1

Reviewer 1 Report

The paper reports on work that is well within the normal parameters of this type of ice nucleation studies. The comparison of different strains of Fusarium is a reasonable goal and the results show it to have led to useful results. The comments that follow focus on the ice nucleation measurements and the representation of the results.

Stepwise cooling of the sample, as done in this work, has the disadvantage of combining steady cooling and constant temperature periods, each of which, separately, has recieved considerable analysis in the literature but the combination has not. Thus, the resuults are somewhat more difficult to compare with much of the literature. However, specially in the study of biological INPs, the stepwise cooling method has been used in a fair number of papers because it is a convenient arrangement and doesn't demand fast detection of freezing events. Also, time is a secondry factor in determining the temperature at which freezing is expected for a given INP (cf. Section 3.2.2. in Vali, 2014) and the analyses in terms of nucleus spectra neglects that secondary facoty. Hence, for the study here reported this is not of major concern and the intercomarisons are affected to a less significant degree. It is suggested that the paper include a brief reference to this issue in order to avoid possible misinterpretations in comparisons of these results with other work. 

A general comment is necessary regarding the way INA is talked about in the paper. Phrases like " ... strains showed INA above -12°C" (lines 74-75) are imprecise statements, As is well known, and as the results shown in the paper also demontrate, there is a temperature spectrum associated with any sample, so a single temperature is not meaningful. Even though threshold temperatures are mentioned in some published works, that is not a good prectice because of its loose definition. The paper could be easier to follow if the concentration per gram was given a somple symbol (K is used in many similar contexts in other papers; with K(T) to denote the spectrum and K-10, por example, referring to the value of K at -10°C). The ordinate of Figs. 1 to 3 could then be indicated aslg(K). This would also help in avoiding vague terms like "strength of INA", "IN/g" and other less clearly defined temrs. The paper should be reviewed with this comment in mind

line, figure or section reference

30 The homogenous nucleation temperature is not such a fixed value as indicated here. Actual temperature of homogeneous freezing dependes on the sampe volume in a well-known fashion. It is usually taken to be taking place somewhere below -36°C. There are small difference among different studies. Furthermore, homogeneous freezing is not defined by a temperature but by the absence of any impurities in the water.

96-97 Is there any melting point depression due to the solution effect arising from the culture medium?

Fig. 1 Is the ordinate scale given with natural logarith or base 10 log? 

3.1 The number of freezing event at -6°C is probably too small to allow assigning significance to the differences among the samples. On the other hand, the 13 samples with greater activity have remarkably similar curves in Fig. 1. So, differences within this group could be more clearly defined by the concentration of INPs at some temperature , say <-8°C, or at -10°C.Something close to this is done in line 262, 283 and other places. The differences span over an order of magnitude and may have some qualitative factors associated with them. The paragraph (lines 227-232) should be re-thought.

255-271 This is an interesting result. One question related to this is the possible effect of time taken to do the series of tests. Is the initial sample stable over this period?

528 " .. strains had INA." is understandable in the context of what heas been described in the preceding parts of the paper, but taken at face value it is a loose statement, that would be good to make more conscise, especially since it is the first sentence of Conclusions. 

Reviewer 2 Report

This manuscript presents a large body of information on a variation of ice nucleation activity (INA) among 15 Fusarium avenaceum strains and recognition of putative genes involved in the formation of ice nucleating particles (INPs). The authors assessed the strains' INA at different temperatures, characterized their INPs in samples prepared under different conditions, and analyzed genomic and transcriptomic datasets of all tested strains in a technically robust manner. They revealed manifold candidate genes that may deepen insight into Fusarium INA although they remain to be functionally characterized. The manuscript is well written and will benefit the society with no doubt. Therefore, I recommend it to be accepted for publication in JoF in priority after a minor revision.

Minor suggestions:

1.        Please italicize some fungal names not italicized in the text.

2.        In Materials and methods, I guess hat -1 and -5 in 10-1 to 10-5 dilutions must be superscripts.  

3.        In Figures 1 to 3, different symbols (shape and solid/open) and colors could be more distinguishable for the data points of different strains or treatments

4.        Change 30kDa to 30 kDa (insert a space) throughout the text.

5.        For statistic significance, change p-value to p (italicized) throughout the text.

6.        In Figure 5A, setting the minimum of X-axis to -15 at an interval of 5 would improve the graph.

7.   Try you best to make a revision more concise.
